# Introduced Spiders in Panama: Species Distributions and New Records

**DOI:** 10.3390/biology14010004

**Published:** 2024-12-24

**Authors:** Daniel Murcia-Moreno, Dumas Gálvez

**Affiliations:** 1Smithsonian Tropical Research Institute, Ciudad de Panamá 0843-03092, Panama; daniel.murcia@up.ac.pa; 2Estación Científica Coiba, Coiba AIP, Calle Gustavo Lara, Bld. 145B, Clayton 0843-01853, Panama; 3Programa Centroamericano de Maestría en Entomología, Universidad de Panamá, Estafeta Universitaria, Avenida Simón Bolívar, Ciudad de Panamá 0824, Panama

**Keywords:** biological invasion, exotic species, Central America, neotropics

## Abstract

Organisms in nature are often not restricted to a single geographical location and are able to disperse to new areas. However, humans have provided means for many species to disperse over much larger distances (e.g., ships, planes) than those they would normally reach by natural means. Introduced species often can have negative effects for native organisms and local economies (known as invasive species) and for this reason a first step is to quantify the number of introduced species in a particular region. Although arachnids are a group that has been extensively studied in terms of invasive species, some countries lack information on this issue, which is the case for Panama. Here we provide the first list of introduced spiders in Panama. By reviewing scientific literature, museum collections, online databases and fieldwork, we identified 31 species of introduced spiders in the country and six of those were identified by this study. Although none of the species seem to generate negative effects for local ecosystems, more research is needed to evaluate any potential effects to native species. Similarly, monitoring policies should consider spiders as a group of organisms with potential risk for local ecosystems.

## 1. Introduction

The dispersal of organisms is essential for the distribution of life on Earth, it plays a critical role for the maintenance of biodiversity, contributing to the strengthening of diverse forms and functions in living beings [1]. However, the arrival of exotic species may impose new challenges for native species, like new competitors and diseases, predation and habitat alteration, among others [2,3,4,5,6]. Expansion in the distribution ranges of species between continents is mainly driven by human activities, and it is experiencing an increase in recent decades [1,7,8]. Introduced non-native species to a particular ecosystem that can cause or are likely to cause harm to the economy, environment or human health are known as invasive species [9]. In that line, some introduced species may not acquire the status of invasive since they do not seem to generate a clear problem [10].

Globally, most research on invasive species has focused on plants, vertebrates, and several groups of invertebrates, including spiders [7,8,11,12]. However, it is crucial to recognize that invasive spiders can have a significant impact on ecosystems due to their ability to reach high densities in both natural and urban environments, potentially affecting native species diversity [7,13]. In addition, some of these invasive spiders also have medical relevance, as their bites can have potentially lethal consequences [14,15,16].

Panama is recognized for its rich biological diversity and biological invasions have received continued attention. Over the years, a variety of invasive taxa have been reported, including plants, birds, fish, reptiles, amphibians and invertebrates [17,18,19,20,21,22,23,24,25,26]. The introduction of these species to Panama is the result of multiple activities like pest control, promotion of agriculture and aquaculture, exotic pet trade and accidental introductions due to human activity. Unfortunately, these introductions, instead of generating benefits, have given rise to significant problems in terms of economic and ecological damages [26,27,28]. In spite of those efforts, invasive spider species has received little attention in government reports [29] or scientific studies [26].

In the country, the first list of spider species by Nentwig [30] registered 1223 species, which currently reaches approximately 1240 species (Murcia-Moreno, unpubl. data). However, the proportion of introduced species within this figure remains unknown. Therefore, here we present a list of introduced species resulting from a review of scientific literature, review of a museum collection, review of available databases and opportunistic sampling. Our study provides a complement to a recent work by Rodríguez-Gavilanes et al. [26] that listed invasive species of Chordates and Arthropods in Panama but it did not include spiders in their report. Moreover, the present work provides basic information on the diversity and distribution of introduced spider species in the country. Although further work is required to decipher potential ecological or economic costs imposed by these species, we assumed an invasive status for many of them, given their wide distribution and long establishment in the country.

## 2. Materials and Methods

This study was carried out from 2022–2024 and consisted of an extensive bibliographic review of scientific articles and online databases: (1) World Spider Catalog [31]; (2) The Global Biodiversity Information Facility [32] and (3) iNaturalist [33]. We used the list of species provided by Nentwig [30] as the main guide to check for introduced species. None of the most recent reviews on biological invasions in Panama provided information on introduced spiders [26,29].

We also used Web of Science (WOS: all databases) to search for articles on introduced spider species in Panama. We performed independent searches with the terms ‘invasive species’, ‘invasive spider’, ‘exotic spider’, ‘especie invasora’ and ‘especie exotica’. Each of these searches was refined with the term ‘Panama’ (Appendix A).

Our field work was based on an unsystematic sampling of sites visited during 2022–2024 during both dry and rainy seasons. The spiders that we sampled were opportunistically collected from visits to multiple sites that included natural and anthropogenic habitats like ports, buildings in urban and rural areas, green areas in urban areas, protected areas and tourist sites. We collected spiders by means of forceps or aspirators and were placed into 70% ethanol. Collections were made under the permit 0033-2021 (Ministry of Environment). Finally, we reviewed specimens from the Invertebrate Museum of the University of Panama (MIUP).

From the review and visits to the MIUP, we obtained historical records (first time report) and distribution data for all species. We considered species as introduced or cosmopolitan following the taxonomic nomenclature of the World Spider Catalog [31]. We identified collected spiders by morphological characteristics, by means of a stereomicroscope (Nikon SMZ 800, Tokyo, Japan). We confirmed identifications following published guides in scientific articles, books and websites. Samples are stored in the laboratory of Dumas Gálvez at the Central American Masters in Entomology, University of Panama. All samples are freely available.

## 3. Results

Our WOS searches did not generate documents of which we were not already aware (Appendix A). Here, we report 31 species of introduced spiders to Panama, belonging to 10 families and 25 genera (Table 1). The families with the highest number of representative species were Theridiidae (7 spp.), Oonopidae (7 spp.), Araneidae (4 spp.) and Pholcidae (4 spp.) (Table 1, Figure 1). Species from Asia seem to be the most common introduced species (Table 1). We also report six new introduced species to the country: *Crossopriza lyoni* (Blackwall, 1867) (Figure 2), *Cyrtophora citricola* (Forsskål, 1775) (Figure 3), *Neoscona adianta* (Walckenaer, 1802), *Parasteatoda tepidariorum* (C. L. Koch, 1841), *Steatoda grossa* (C. L. Koch, 1838) and *Theridion melanostictum* (O. Pickard-Cambridge, 1876) (Figure 4, Table 1 and Table 2 and Appendix A).

We found evidence from the museum collection and our sampling that some species have expanded their distribution in the country (Table 2) as compared to the historic data (Table 1). We also report new location records for most of the species; for instance, *Cithaeron praedonius* (O. Pickard-Cambridge, 1872) (Figure 2, Table 2), *Labahitha marginata* (Kishida, 1936) (Figure 5, Table 2) and *Micropholcus fauroti* (Simon, 1887) (Figure 6, Table 2). Moreover, rooted on a presence/absence evaluation per province, *Cyrtophora citricola* (all provinces), *Heteropoda venatoria* (Linnaeus, 1767), *Labahitha marginata* and *Physocyclus globosus* (Taczanowski, 1874) seem to be the most widespread species in the country (Figure 1, Table 1 and Table 2 and Appendix A). Overall, the provinces of Panama, Panamá Oeste, Colón and Chiriquí seem to have received the largest numbers of introduced species and widest distributions locally (Table 2 and Table 3 and Appendix A).

We also present data on species that could be present in Panama, given their presence in the two neighboring countries (Costa Rica and Colombia), but have not yet been reported, as well as records that required confirmation due to insufficient evidence (Table 3 and Appendix A).

## 4. Discussion

The introduced spiders in Panama present a diverse and complex panorama in terms of their distribution and origin. Most of the species recorded belonged to the families Theridiidae, Oonopidae, Araneidae and Pholcidae, in comparison to regions like South America, where the most common introduced species belong to Oonopidae, Theridiidae, Salticidae and Gnaphosidae [92] or in Europe, where Pholcidae and Theridiidae contain a large proportion of the introduced species [7]. The dominance of families such as Theridiidae, Pholcidae, Oonopidae and Araneidae could be related to their morphological size, the similarity of their climatic niches with those of their place of origin and their ability to adapt to microclimatic changes (e.g., anthropogenic factors). These families include numerous species well adapted to dry niches in their areas of origin, which facilitates their survival especially in urban environments, which is the case in for instance in Europe [100]. Importantly, future work with molecular techniques such as DNA barcoding could improve the accuracy on the detection of introduced species and establish the point of origin.

Although it is difficult to establish the origin of the introduced species, our results suggest that species from Asia are the most common invaders in Panama. However, certainty of the invasion origin of some species is low, given that they could come from a location that is not necessarily their site of origin (e.g., *B. parumpunctata*, *O. deserticola*) [92]. Still, spider introductions in Panama could be partially explained by the constant traffic through the Panama Canal which allows the transfer of organism by ships [101] and the fact that the Atlantic—Pacific route involving Asia is a major traffic route through the canal [102]. This hypothesis is supported by the fact that provinces around the canal (Panama, Panamá Oeste and Colón) showed the largest proportion of introduced species. In line with our results, Rodríguez-Gavilanes et al. [26] found the highest number of invasive taxa for the province of Panama. Another likely entry point is the transport network [103] connection between the Chiriquí province and Costa Rica, and this province ranks fourth in the proportion of introduced species recorded, in line with this idea. Also supporting this idea is the fact that the other province that borders Costa Rica (Bocas del Toro), but with significantly less traffic, show relatively low number of introduced species, as well as Darién with no road connections with Colombia. Still, there is the possibility that these figures are the result of biased sampling in each province.

Understanding whether sampling bias is an issue to understand introduced species distribution is important because it can shed some light also on whether these species can naturally disappear in the sites where they manage to invade and establish. For instance, *Latrodectus mactans* and *Artema atlanta* have not been reported in almost a century [30,54]. Thus, if those species have truly disappeared then which factors are responsible? Related to this, more work is needed to understand whether there is some correlation between habitat type and the likelihood of establishment by introduced species [104]. For instance, the apparent most common introduced species in Panama, *Cyrtophora citricola*, has been able to occupy a novel set of abiotic conditions in the Americas, which are not present in their native range of distribution [8]. Still, most of the introduced species in Panama come from warm regions such as Africa, Asia and the Mediterranean (Table 1), which suggests that they find favorable similar abiotic conditions in the country.

The distribution pattern can also provide estimation on colonization dates. A study from Chile suggests that residence time is a determining factor in predicting the invasive success and range size of introduced spiders. Species with more than 100 years in the country have shown significant range expansions [105] and other studies have found a positive relationship between residence time and invasion success [106,107]. Based on this hypothesis, species such as *C. citricola*, *L. marginata* and *M. bivittatus* may have established in Panama a long time ago and be among the first species to arrive, which could explain their wide distribution range.

New distribution ranges that include Panama are also expected for introduced spiders that are currently present in the two neighboring countries, Costa Rica and Colombia. It is likely that there are introduced spider species in Panama that have not yet been confirmed, such as *Tegenaria domestica* and *Steatoda nobilis*, in addition to those that require more evidence, such as *Oecobius navus* and *Pholcus phalangioides*. Quintero [64] mentioned the presence of *O. navus* in the country, but did not provide references or collection data. We found no historical or current evidence to support its presence in Panama and the specimens that we have collected correspond to *Oecobius cocinnus*, a native and synanthropic species. Although we did not collect specimens of *P. phalangioides* in this study, we identified a previous record probably from Panama. However, the associated coordinates are not reliable, and the date of collection is unknown. This record was identified by Pholcidae expert Hubert Bernhard [97]. The possibility that other unconfirmed species are in the process of becoming established in the country is feasible. While some introduced species are known to form populations quickly, others require more time and studies to determine if they can maintain sustainable populations in the new ecosystem [92].

Once introduced species establish in those new locations, it remains unknown whether some of these species could be biologically controlled [108] or whether they could provide ecological services [109]. Moreover, further ecological work is required to understand whether some of these introduced species pose a risk [110] or even benefits [111] for native species, and how their presence may influence local ecosystems with potential cascading effects [112] or interacting effects [4]. For example, *C. citricola* webs can interfere with plant growth in agricultural systems, possibly resulting in a lower yield [113] and pollination success, which would make this species a true invasive species. On the other hand, species like *L. mactans* and *L. geometricus* could pose a health risk for humans and animals due to their potentially risky venoms [105,107].

There have not been studies aimed at understanding the impacts of these introduced species in Panama, either related to ecological or economics effects. However, based on our experience and trends observed in the literature for other invasive spider species [7], we estimate that their overall impact is low. Perhaps one of the species that possess the highest economical risk for coffee plantations is the most widespread, *C. citricola*, as reported in Colombia [113], but this does not seem the case in Panama. Regarding monitoring policies, the Panamanian government have traditionally implemented policies for monitoring arthropod species of risk in entry points of the country, for example, species that represent a threat for agriculture [114]. Unfortunately, there has not been monitoring programs for other taxa, like spiders; however, recent policies are targeted to cope with the issue of invasive species in general [34]. Our work highlights the fact that spiders are part of the taxa that can easily establish and spread in the country, which calls for attention of the monitoring policies and potentially control strategies.

## 5. Conclusions

Our work has identified a complex scenery for introduced spiders in Panama, originating from different geographic regions, which may be driven by multiple factors such as international trade, transport and environmental change. The new records of introduced spiders expand the knowledge of Panama’s biodiversity and call for a strict monitoring of their distribution and potential ecological impacts, which includes their interaction with native organisms, and long-term impacts. Our work updates the list of introduced animal species in the country to 172, and arachnids represent 19% of the animal species recorded. Future efforts should focus on monitoring new introductions, validating historical records, and establishing policies that minimize the ecological and health risks associated with these species.

## Figures and Tables

**Figure 1 biology-14-00004-f001:**
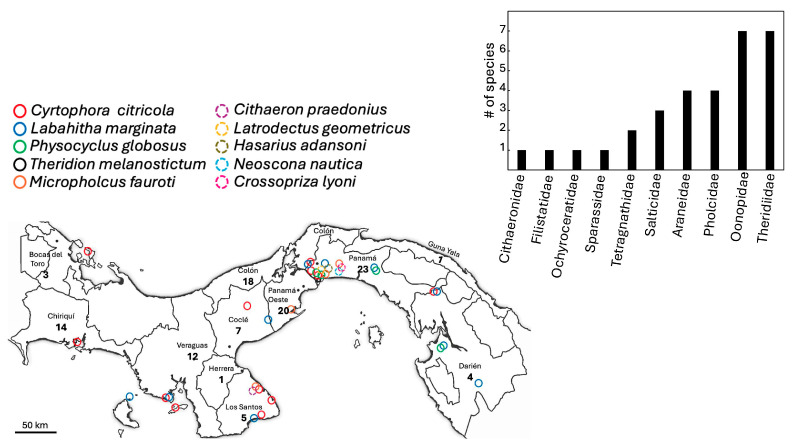
New records of introduced species based on collected individuals from different locations in Panama, and the total number of invasive species per province. Box shows the number of species per family in the country.

**Figure 2 biology-14-00004-f002:**
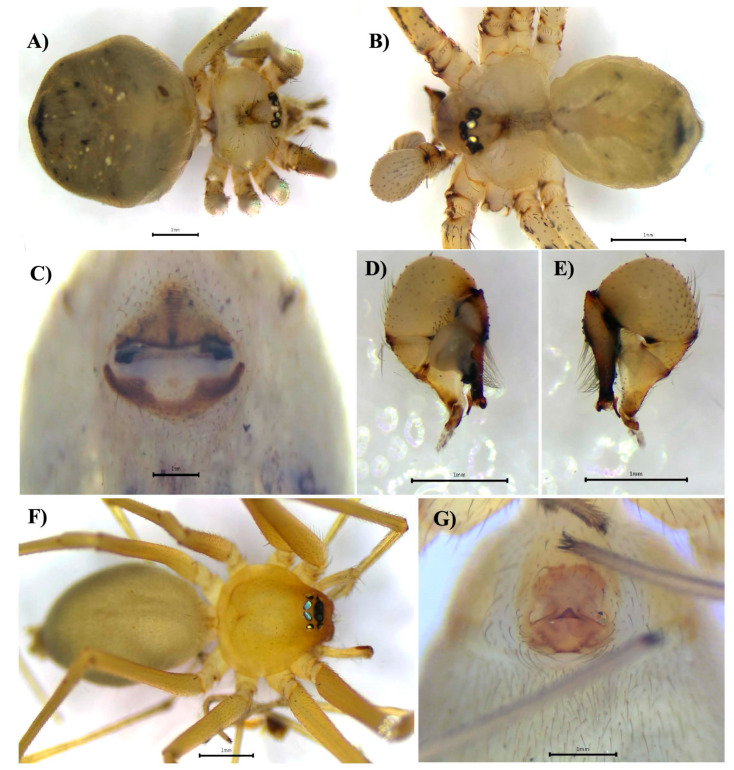
*Crossopriza lyoni*, female, dorsal view (**A**). Male, dorsal view (**B**). Female epigynum, ventral view (**C**). Male left palp, prolateral view (**D**), retrolateral view (**E**). *Cithaeron praedonius*, female, dorsal view (**F**). Epigynum, ventral view (**G**).

**Figure 3 biology-14-00004-f003:**
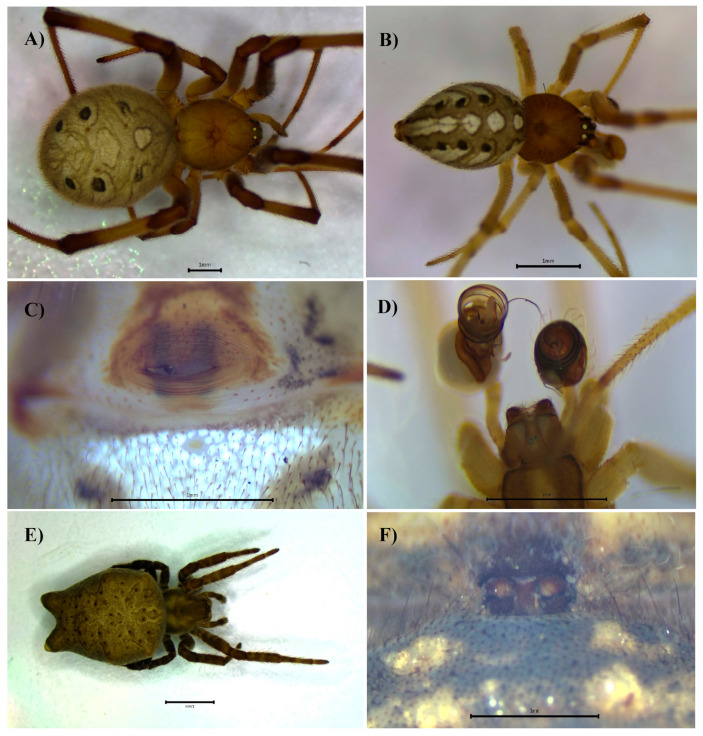
*Latrodectus geometricus*, female, dorsal view (**A**). Male, dorsal view (**B**). Female epigynum, ventral view (**C**). Male palps, ventral view (**D**). *Cyrtophora citricola*, female, dorsal view (**E**). Epigynum, ventral view (**F**).

**Figure 4 biology-14-00004-f004:**
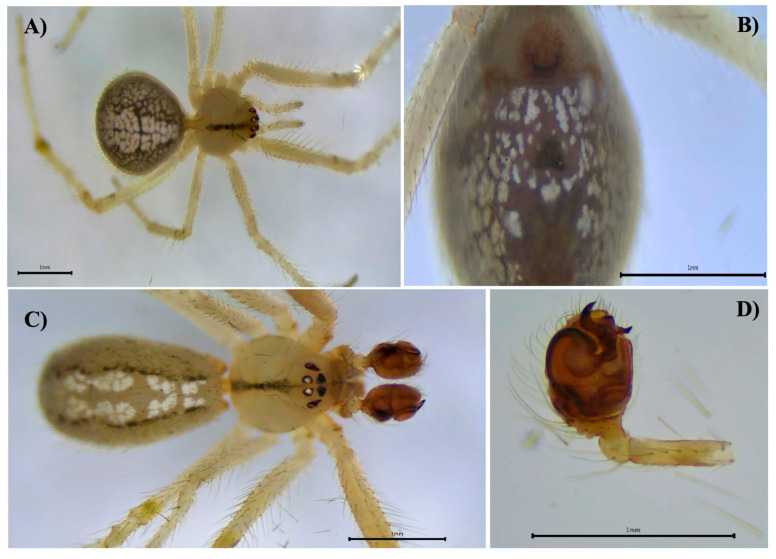
*Theridion melanostictum*, female, dorsal view (**A**). Female epigynum, ventral view (**B**). Male, dorsal view (**C**). Male right palp, ventral view (**D**).

**Figure 5 biology-14-00004-f005:**
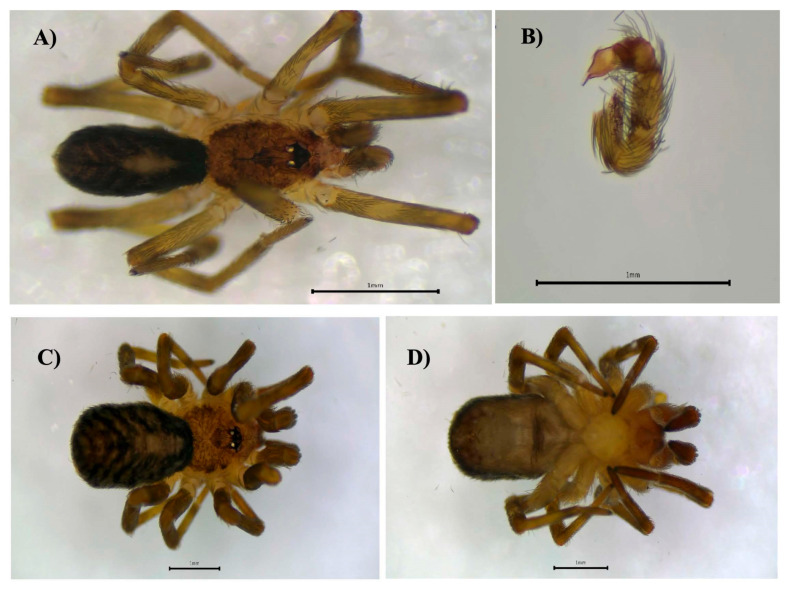
*Labahitha marginata*, male, dorsal view (**A**). Male right palp, retrolateral view (**B**). *Labahitha marginata*, female, dorsal view (**C**). Female, ventral view (**D**).

**Figure 6 biology-14-00004-f006:**
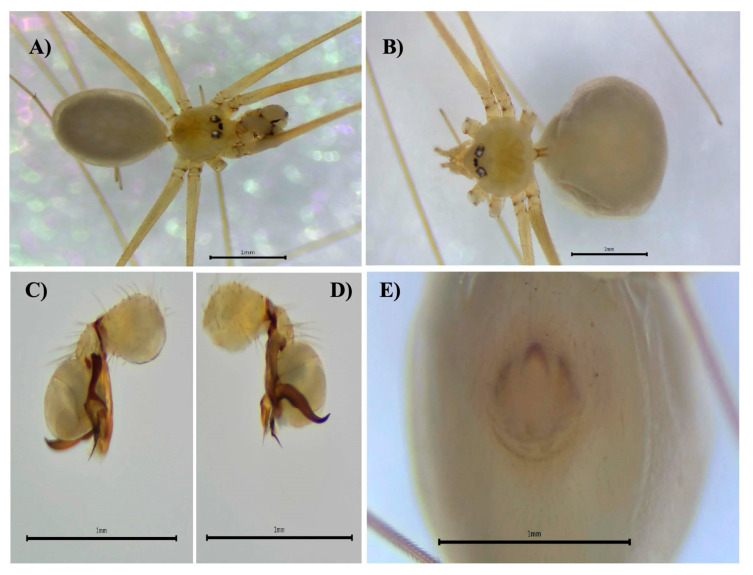
*Micropholcus fauroti*, male, dorsal view (**A**). Female, dorsal view (**B**). Male right palp, prolateral view (**C**), retrolateral view (**D**). Female epigynum, ventral view (**E**).

**Table 1 biology-14-00004-t001:** List of first-time reports of introduced spider species in Panama.

Species	Area of Origin	First Record	Distribution	Reference
**Araneidae**				
*Cyrtophora citricola* (Forsskål, 1775)	Southern Europe, Africa, Middle East, Asia (Pakistan, India, China, Japan).	this work		
*Gea heptagon* (Hentz, 1850)	Pacific Is, Australia.	1929	Panamá Oeste, Panama	[30,34,35]
*Neoscona adianta* (Walckenaer, 1802)	Europe, North Africa, Asia.	this work		
*Neoscona nautica* (L. Koch, 1875)	Asia, Pacific Is.	1885	Bocas del Toro, Chiriquí, Panamá Oeste, Colón	[30,36,37,38,39,40,41,42]
**Cithaeronidae**				
*Cithaeron praedonius* (O. Pickard-Cambridge, 1872)	North Africa, Greece, Cyprus, Turkey, Middle East to India, Malaysia	2024	Panamá Oeste	[31,43]
**Filistatidae**				
*Labahitha marginata* (Kishida, 1936)	Taiwan, Philippines, Papua New Guinea, Pacifi Is.	2022	Chiriquí, Veraguas	[44,45]
**Ochyroceratidae**				
*Theotima minutissima* (Petrunkevitch, 1929)	Tropical Asia	1995	Panamá Oeste	[46]
**Oonopidae**				
*Brignolia parumpunctata* (Simon, 1893)	Asia (Pakistan, India, Sri Lanka, China (Yongxing Is. = Woody Is.), Philippines, Indonesia (Sulawesi, Banda Is.).	1951	Panamá Oeste, Panamá, Colón	[30,47,48,49]
*Ischnothyreus peltifer* (Simon, 1892)	Tropical Asia	1951	Los Santos, Panamá Oeste, Panamá, Colón	[30,47,48,50]
*Ischnothyreus velox* (Jackson, 1908)	Tropical Asia	2012	Panamá, Colón	[50]
*Opopaea apicalis* (Simon, 1893)	Asia (China (Yongxing Is. = Woody Is.), Thailand, Indonesia, Philippines).	2009	Panamá Oeste, Panamá, Colón	[51]
*Opopaea deserticola* (Simon, 1892)	Southeast Asia	1951	Panamá Oeste, Panamá, Colón	[30,47,51]
*Triaeris stenaspis* (Simon, 1892)	Africa	1938	Bocas del Toro, Chiriquí, Coclé, Panamá Oeste, Panamá, Colón	[30,47,48,50,52,53]
*Xestaspis parmata* (Thorell, 1890)	Southeast Asia	2019	no data	[31]
**Pholcidae**				
*Artema atlanta* (Walckenaer, 1837)	North Africa, Middle East	1925	Colón	[30,54]
*Crossopriza lyoni* (Blackwall, 1867)	Probably native to Africa and/or Asia	this work		
*Micropholcus fauroti* (Simon, 1887)	Temperate Asia	1996	Panamá Oeste	[55,56]
*Physocyclus globosus* (Taczanowski, 1874)	North America	1925	Chiriquí, Veraguas, Coclé, Panamá Oeste, Panamá, Colón	[30,54,56,57,58,59]
**Salticidae**				
*Hasarius adansoni* (Audouin, 1826)	Africa, Middle East	1925	Chiriquí, Panamá	[30,54,60]
*Menemerus bivittatus* (Dufour, 1831)	Africa	1901	Chiriquí, Panamá Oeste, Panamá, Colón	[30,35,54,57,61,62]
*Plexippus paykulli* (Audouin, 1826)	Africa	1929	Panamá Oeste, Panamá, Colón	[30,35,57,61,63]
**Sparassidae**				
*Heteropoda venatoria* (Linnaeus, 1767)	Tropical Asia	1936	Veraguas, Coclé, Panamá Oeste, Panamá, Colón	[64,65,66]
**Tetragnathidae**				
*Tetragnatha nitens* (Audouin, 1826)	Tropical and subtropical Asia	1925	Chiriquí, Coclé, Panamá Oeste, Colón	[30,54,67]
*Tetragnatha vermiformis* (Emerton, 1884)	Temperate and tropical Asia	1981	Panamá Oeste	[30,68,69]
**Theridiidae**				
*Latrodectus geometricus* (C.L. Koch, 1841)	Africa	1959	Panamá, Colón	[70,71]
*Latrodectus mactans* (Fabricius, 1775)	Probably native to North America only.	1902	Chiriquí, Veraguas, Panamá Oeste	[30,38,54,62,70]
*Meotipa pulcherrima* (Mello-Leitão, 1917)	East Asia, Papua New Guinea, Pacific Is.	1936	Chiriquí, Veraguas, Panamá Oeste, Panamá, Colón	[30,57,64,72,73,74]
*Parasteatoda tepidariorum* (C.L. Koch, 1841)	Asia	this work		
*Steatoda erigoniformis* (O. Pickard-Cambridge, 1872)	East Mediterranean, Middle East, Caucasus, India, East Asia	1957	Veraguas, Panamá, Colón	[30,64,75,76]
*Steatoda grossa* (C. L. Koch, 1838)	Europe, Turkey, Russia (Europe to Far East), Caucasus, Iran, Kazakhstan, Central Asia, China, Korea, Japan	this work		
*Theridion melanostictum* (O. Pickard-Cambridge, 1876)	Macaronesia, Mediterranean to Egypt, India, Central Asia, China, Japan	this work		

**Table 2 biology-14-00004-t002:** List of introduced spider species, with new data distribution from collection and databases review. * literature review.

Species	Location	Basis of Record	Reference
**Araneidae**			
*Cyrtophora citricola* (Forsskål, 1775)	Bocas del Toro, Chiriquí, Veraguas, Herrera, Los Santos, Coclé, Panamá Oeste, Panamá, Colón, Comarca Kuna Yala, Darien	Collected specimen, Inaturalist	[77]
*Gea heptagon* (Hentz, 1850)	Panamá Oeste, Panamá	MIUP Collection	
*Neoscona adianta* (Walckenaer, 1802)	Chiriquí	Gbif database	[78]
*Neoscona nautica* (L. Koch, 1875)	Bocas del Toro, Chiriquí, Veraguas, Panamá Oeste, Panamá	MIUP Collection, Inaturalist, Collected specimen	[42]
**Cithaeronidae**			
*Cithaeron praedonius* (O. Pickard-Cambridge, 1872)	Veraguas, Los Santos, Panamá	Collected specimen, Inaturalist	[79]
**Filistatidae**			
*Labahitha marginata* (Kishida, 1936)	Chiriquí, Veraguas *, Los Santos, Coclé, Panamá, Colón, Darien	Collected specimen	[45]
**Oonopidae**			
*Ischnothyreus peltifer* (Simon, 1892)	Los Santos *	MIUP Collection	[50]
*Opopaea deserticola* (Simon, 1892)	Panamá	MIUP Collection	
*Xestaspis parmata* (Thorell, 1890)	Panamá Oeste, Panamá, Colón	Gbif database	[80]
**Pholcidae**			
*Crossopriza lyoni* (Blackwall, 1867)	Panamá	Collected specimen	
*Micropholcus fauroti* (Simon, 1887)	Los Santos, Panamá	Collected specimen	
*Physocyclus globosus* (Taczanowski, 1874)	Panamá, Colón, Darien	Collected specimen	
*Crossopriza lyoni* (Blackwall, 1867)	Panamá	Collected specimen	
** *Salticidae* **			
*Hasarius adansoni* (Audouin, 1826)	Veraguas, Coclé, Panamá, Colón	MIUP Collection, Collected specimen, Inaturalist	[81]
*Menemerus bivittatus* (Dufour, 1831)	Bocas del Toro, Chiriquí, Veraguas, Los Santos, Herrera, Coclé, Panamá Oeste, Panamá, Colón	MIUP Collection, Collected specimen, Inaturalist	[82]
*Plexippus paykulli* (Audouin, 1826)	Veraguas, Panamá Oeste, Panamá	MIUP Collection, Collected specimen, Inaturalist	[83]
**Sparassidae**			
*Heteropoda venatoria* (Linnaeus, 1767)	Chiriquí, Veraguas, Panamá Oeste, Panamá, Comarca Kuna Yala, Darien	Inaturalist	[84]
**Tetragnathidae**			
*Tetragnatha nitens* (Audouin, 1826)	Panamá	MIUP Collection	
**Theridiidae**			
*Latrodectus geometricus* (C.L. Koch, 1841)	Panamá	MIUP Collection, Collected specimen	
*Parasteatoda tepidariorum* (C.L. Koch, 1841)	Chiriquí	Gbif database, Inaturalist	[85,86]
*Steatoda grossa* (C.L. Koch, 1838)	Chiriquí, Veraguas	Inaturalist	[87]
*Theridion melanostictum* (O. Pickard-Cambridge, 1876)	Panamá Oeste, Panamá, Colón	Gbif database, Collected specimen	[88]

**Table 3 biology-14-00004-t003:** Putative introduced spider species or with the potential for introduction in Panama.

Species	Location	Basis of Record	Reference
**Agelenidae**			
*Tegenaria domestica* (Clerck, 1757)	not reported	Costa Rica, Colombia	[62,89,90]
*Tegenaria pagana* (C.L. Koch, 1840)	not reported	Colombia	[90,91]
**Araneidae**			
*Neoscona moreli* (Vinson, 1863)	not reported	Colombia	[31,40,92]
*Nephilingis cruentata* (Fabricius, 1775)	not reported	Colombia	[31,90,92,93]
**Gnaphosidae**			
*Marinarozelotes kulczynskii* (Bösenberg, 1902)	not reported	Colombia	[31,90,92,94]
*Urozelotes rusticus* (Fabricius, 1775)	not reported	Colombia	[90]
**Oecobiidae**			
*Oecobius navus* (Blackwall, 1859)	to be confirmed	Colombia	[64,90,92,95]
**Oonopidae**			
*Opopaea concolor* (Blackwall, 1859)	not reported	Costa Rica, Colombia	[51,92]
**Pholcidae**			
*Pholcus phalangioides* (Fuesslin, 1775)	to be confirmed	Costa Rica	[96,97]
*Smeringopus pallidus* (Blackwall, 1858)	not reported	Costa Rica	[56,96,98]
**Theridiidae**			
*Steatoda nobilis* (Thorell, 1875)	not reported	Colombia	[31,90,92,99]

## Data Availability

All data supporting this manuscript is available as Appendix A.

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
