# Peer review of "Introduced Spiders in Panama: Species Distributions and New Records"

_biology, 2024, doi:10.3390/biology14010004_

Round 1
Reviewer 1 Report
Comments and Suggestions for Authors
The study described is a well-structured investigation into the presence of invasive spider species in Panama. It is thorough in its approach, combining extensive bibliographic review, field sampling, and museum specimen analysis to achieve a comprehensive understanding of spider biodiversity and invasive species in the region. Below is a detailed review of its methodology and strengths:
While the study identifies historical records, it would benefit from more detailed comparative analysis of changes in species distributions over time.
Though implied, ethical clearances for sampling in protected areas and proper handling of biodiversity could be explicitly stated.
While the methodology is well-described, the actual data (e.g., number of species identified, distribution metrics) is not included in this excerpt. Supplementary data visualization in the form of charts or tables (e.g., Table S1) would enhance clarity.
Morphological identification is robust but could be supplemented with molecular techniques such as DNA barcoding to improve accuracy, especially for cryptic species.
The text notes the absence of invasive spider records in recent reviews but does not elaborate on whether the study itself identified any invasive species. This clarification is crucial to establish its contribution to understanding biological invasions.
The study presents findings on introduced spider species in Panama based on extensive research combining museum records, field sampling, and bibliographic review. It highlights the diversity, origins, distribution, and potential future trends of invasive spiders in the region. The study is scientifically valuable, offering new data on invasive arachnid fauna and filling gaps in the literature.
However
1. Although museum and sampling data are used, the exact criteria for the "presence/absence evaluation" across provinces could be elaborated for reproducibility.
2. The records requiring confirmation (Table 3 and S4) are mentioned but not elaborated upon in the text. A brief discussion of why these records remain uncertain would provide clarity.
3. The dominance of Theridiidae and other families, as well as the reasons for their success, could be explored further (e.g., life history traits, adaptability to anthropogenic habitats).
4. The text references multiple tables and figures, but without visualizing these data, the interpretation of results is less accessible. Including summaries or interpretations of these figures would enhance the narrative.
5. Discuss the impact of these species on native ecosystems and human activities in Panama
6. Investigate how these species are entering the country (e.g., trade, tourism) to inform management strategies
7/Suggest long-term monitoring programs to track the spread and potential impact of the identified species.
Author Response
Reviewer 1
The study described is a well-structured investigation into the presence of invasive spider species in Panama. It is thorough in its approach, combining extensive bibliographic review, field sampling, and museum specimen analysis to achieve a comprehensive understanding of spider biodiversity and invasive species in the region. Below is a detailed review of its methodology and strengths:
While the study identifies historical records,it would benefit from more detailed comparative analysis of changes in species distributions over time.
R: This is the only comment out of the three reviewers that we cannot really achieve to adopt. There are not enough studies or samplings along time to make comparisons over time. However, our discussion make a little description of few species that established a long time ago.
Though implied, ethical clearances for sampling in protected areas and proper handling of biodiversity could be explicitly stated.
R: Added to the methods
While the methodology is well-described, the actual data (e.g., number of species identified, distribution metrics) is not included in this excerpt.
R: We have modified the map to show number of species per province and number of families in the country.
Supplementary data visualization in the form of charts or tables (e.g., Table S1) would enhance clarity.
R: We have modified some of the tables for clarification.
Morphological identification is robust but could be supplemented with molecular techniques such as DNA barcoding to improve accuracy, especially for cryptic species.
R: we added this to the discussion as future work.
The text notes the absence of invasive spider records in recent reviews but does not elaborate on whether the study itself identified any invasive species. This clarification is crucial to establish its contribution to understanding biological invasions.
R: This was already mentioned in the results but we made it more clear by adding it to the abstract.
The study presents findings on introduced spider species in Panama based on extensive research combining museum records, field sampling, and bibliographic review. It highlights the diversity, origins, distribution, and potential future trends of invasive spiders in the region. The study is scientifically valuable, offering new data on invasive arachnid fauna and filling gaps in the literature.
However
- Although museum and sampling data are used, the exact criteria for the "presence/absence evaluation" across provinces could be elaborated for reproducibility.
R: This evaluation was done based on Table 1 and 2. We have added more details for clarification.
- The records requiring confirmation (Table 3 and S4) are mentioned but not elaborated upon in the text. A brief discussion of why these records remain uncertain would provide clarity.
R: Added
- The dominance of Theridiidaeand other families, as well as the reasons for their success, could be explored further (e.g., life history traits, adaptability to anthropogenic habitats).
R: added
- The text references multiple tables and figures, but without visualizing these data, the interpretation of results is less accessible. Including summaries or interpretations of these figures would enhance the narrative.
R: we modified the map, to show number of species per province and number of species per family.
- Discuss the impact of these species on native ecosystems and human activities in Panama
R: We have discussed potential effects in the original version, but we have not expanded it.
- Investigate how these species are entering the country (e.g., trade, tourism) to inform management strategies
R: We have discussed this, but we have now expanded a bit more.
7.Suggest long-term monitoring programs to track the spread and potential impact of the identified species
R: Added. Particularly discussed for C. citricola.
Reviewer 2 Report
Comments and Suggestions for Authors
I believe the manuscript has a significant contribution to the field of invasive species research, particularly regarding arachnids and Latin America. The study has a good methodology and fills an important gap on spider knowledge in Panama. However, there are several areas where the manuscript could benefit from additional attention to improve clarity, consistency, and overall rigor. The whole paper needs a clear and consistent definition of “invasive species,” as this term is central to the study's focus. Providing this context early would help avoid confusion, particularly since the terms "invasive," "exotic," and "introduced" are used interchangeably throughout the manuscript. The authors also mention “significant problems” caused by invasive species but do not elaborate on these issues. Explicitly describing these problems would enhance the reader's understanding of the ecological and practical implications of invasive arachnids. The sampling methods need to be further described in greater detail. It is unclear how and where spiders were collected, which habitats were targeted, and during which seasons the sampling occurred. This information is crucial for understanding the scope of the study and its potential limitations. In results, the inconsistency regarding the addition or not of the original authors that describe the species have to be reviewed and add to all of them. Finally, the use of terms like "extinct" or "extinction" to describe invasive species is problematic. These terms carry significant weight and may imply conservation priorities that are not aligned with the context of invasive species.
Overall, the manuscript provides a critical and novel perspective on invasive arachnids, an understudied area of invasion biology. The manuscript also includes minor issues that could be easily addressed with a deep review, to improve overall clarity and precision. I commend the authors for their important contributions and encourage them to refine these points to strengthen the study further and to be accepted. For now, I considered that the paper needs major revisions.
Major:
Introduction:
Line 47: It is important to define what is an invasive species first, since the definition can change depending on the author. For instance, one can consider that not all exotic species are invasive, but all invasive species are exotic. Talking about invasive species here, without context or definition, would make the reader be lost. Please, define it first.
Line 60: Do not let the reader hanging. If you talk about “significant problems” you have to describe what those problems are. What are the problems?
Line 65: Please, check for consistency in terminology regarding “introduced” or “exotic” or “invasive”. These words are correct but they are not always interchangeable. If you want them to be interchangeable, you have to define it in the beginning of your introduction, as the same suggestion for the “invasive” term.
Methods:
Lines 74 and 75: You have to add the references of the World Spider Catalog, GBIF and iNaturalist. Moreover, describe them by their title. For instance, 1) World Spider Catalog (World Spider catalog, 2024); GBIF (GBIF, 2024); etc… You actually proper cited the WSC in the last methods paragraph, hence, do the same here.
Lines 82 to 85: What 70% ethanol means here? I imagine that has something to do with the way that you sampled the spiders, but it is not clear. Because you did not add: how did you collect the spiders? Pitfalls, beating? In which season? Where? Through the whole country? In natural and anthropogenic habitats? Please, describe your whole process of sampling. This is very important because if an exotic species is naturalized you may not sample it depending on the time of the year. Even if your samples were random, you have to properly describe it.
Results:
Line 96: As I said in the introduction, you use “introduced” here. So, without the previous definition, the reader can be confused if the species is invasive or not, even if it is exotic or not. Granted, even a native species, from a country, can be introduced from a habitat to another within the same country. Please, check the definition to maintain a consistent use of terms through the text.
Line 98: Check the misspelling of Oonopidae.
Line 100: Check the misspelling of Neoscona adianta.
Through results text: You have to be consistent with the added names of the author of the species. For instance, you added the names in the main tables but not in all tables of the Supp. Material. At the same time, in the text, you did not put any author name. Please, add the author’s name when you talk about the species for the first time.
Discussion:
Lines 149 to 154: Here you come back to the “invasive” term. This goes to my previous point. You are not being consistent with your terminology. In this same paragraph you use “invasive” and “introduced” as exchangeable terms. They are not. Please, check the definition to maintain a consistent use of terms through the text.
Lines 171 to 175: I do not agree with you when you use the terms “extinct” or “extinction” so lightly for exotic invasive species. First, they are invasive so, they are not in their natural habitat and, supposedly, should not be there either way. Second, using these terms can open an argument about protecting not only invasive spiders but other species that can be more dangerous or harmful to natural habitats or even humans. Hence, there is better terms like “disappear” or “not found anymore” that will give the same message without the weight of extinction. For instance: “… it can shed some light also on whether these species can disappear in the sites where they manage to invade and establish”; or “…if those species are truly not found anymore which are the factors responsible for this disappearance?”
Line 188: Check the misspelling of L. marginata.
Line 197: Check the misspelling of L. mactans.
Conclusion:
Line 211: Landscape is not a proper word to use here, since you did not use any landscape metrics or environmental metrics, besides geographic points, to make your study.
Minor:
Simple summary:
Lines 17 and 18: Change to: “Arachnids are often disregarded in invasive species studies. In this way, here we aim to provide the first list of invasive spiders in Panama.”
Lines 20: Change “field collection” to “field sampling”.
Introduction:
Line 42 to 44: Change to: “However, the arrival of exotic species may impose new challenges for native species, like new competitors and diseases, predation and habitat alteration, among others”. Monoculture and economic damage are not included in the “dispersal of organisms” since it is a human alteration done with intention. Besides, economy is not important for native species, it is only important for humans.
Line 48: Remove the second “as” after spiders and before have. Add a comma after “spiders”.
Line 67: Change “incidental” to “museum”. Check the gap between the end of this sentence and “Our study”.
Line 73: Change “exhaustive” to “extensive”.
Methods:
Line 93: There is a “3” at the end of this line that I imagine that was for the title of the Results.
Author Response
Reviewer 2
The whole paper needs a clear and consistent definition of “invasive species,” as this term is central to the study's focus. Providing this context early would help avoid confusion, particularly since the terms "invasive," "exotic," and "introduced" are used interchangeably throughout the manuscript. The authors also mention “significant problems” caused by invasive species but do not elaborate on these issues. Explicitly describing these problems would enhance the reader's understanding of the ecological and practical implications of invasive arachnids.
R: clarified
The sampling methods need to be further described in greater detail. It is unclear how and where spiders were collected, which habitats were targeted, and during which seasons the sampling occurred. This information is crucial for understanding the scope of the study and its potential limitations.
R: clarified
In results, the inconsistency regarding the addition or not of the original authors that describe the species have to be reviewed and add to all of them.
R: we have added the authority to each species in the results section and also kept them in Table 1.
Finally, the use of terms like "extinct" or "extinction" to describe invasive species is problematic. These terms carry significant weight and may imply conservation priorities that are not aligned with the context of invasive species.
R: modified
Major:
Introduction:
Line 47: It is important to define what is an invasive species first, since the definition can change depending on the author. For instance, one can consider that not all exotic species are invasive, but all invasive species are exotic. Talking about invasive species here, without context or definition, would make the reader be lost. Please, define it first.
R: we have now defined it.
Line 60: Do not let the reader hanging. If you talk about “significant problems” you have to describe what those problems are. What are the problems?
R: modified
Line 65: Please, check for consistency in terminology regarding “introduced” or “exotic” or “invasive”. These words are correct but they are not always interchangeable. If you want them to be interchangeable, you have to define it in the beginning of your introduction, as the same suggestion for the “invasive” term.
R: clarified
Methods:
Lines 74 and 75: You have to add the references of the World Spider Catalog, GBIF and iNaturalist. Moreover, describe them by their title. For instance, 1) World Spider Catalog (World Spider catalog, 2024); GBIF (GBIF, 2024); etc… You actually proper cited the WSC in the last methods paragraph, hence, do the same here.
R: modified
Lines 82 to 85: What 70% ethanol means here? I imagine that has something to do with the way that you sampled the spiders, but it is not clear. Because you did not add: how did you collect the spiders? Pitfalls, beating? In which season? Where? Through the whole country? In natural and anthropogenic habitats? Please, describe your whole process of sampling. This is very important because if an exotic species is naturalized you may not sample it depending on the time of the year. Even if your samples were random, you have to properly describe it.
R: clarified
Results:
Line 96: As I said in the introduction, you use “introduced” here. So, without the previous definition, the reader can be confused if the species is invasive or not, even if it is exotic or not. Granted, even a native species, from a country, can be introduced from a habitat to another within the same country. Please, check the definition to maintain a consistent use of terms through the text.
R: clarified
Line 98: Check the misspelling of Oonopidae.
R: corrected
Line 100: Check the misspelling of Neoscona adianta.
R: corrected
Through results text: You have to be consistent with the added names of the author of the species. For instance, you added the names in the main tables but not in all tables of the Supp. Material. At the same time, in the text, you did not put any author name. Please, add the author’s name when you talk about the species for the first time.
R: Corrected in the main document. For the supplementary material, we reference Table 1 for details on species authorities.
Discussion:
Lines 149 to 154: Here you come back to the “invasive” term. This goes to my previous point. You are not being consistent with your terminology. In this same paragraph you use “invasive” and “introduced” as exchangeable terms. They are not. Please, check the definition to maintain a consistent use of terms through the text.
R: unified throughout the manuscript.
Lines 171 to 175: I do not agree with you when you use the terms “extinct” or “extinction” so lightly for exotic invasive species. First, they are invasive so, they are not in their natural habitat and, supposedly, should not be there either way. Second, using these terms can open an argument about protecting not only invasive spiders but other species that can be more dangerous or harmful to natural habitats or even humans. Hence, there is better terms like “disappear” or “not found anymore” that will give the same message without the weight of extinction. For instance: “… it can shed some light also on whether these species can disappear in the sites where they manage to invade and establish”; or “…if those species are truly not found anymore which are the factors responsible for this disappearance?”
R: corrected
Line 188: Check the misspelling of L. marginata.
R: corrected
Line 197: Check the misspelling of L. mactans.
R: corrected
Conclusion:
Line 211: Landscape is not a proper word to use here, since you did not use any landscape metrics or environmental metrics, besides geographic points, to make your study.
Minor:
Simple summary:
Lines 17 and 18: Change to: “Arachnids are often disregarded in invasive species studies. In this way, here we aim to provide the first list of invasive spiders in Panama.”
R: modified
Lines 20: Change “field collection” to “field sampling”.
R: modified
Introduction:
Line 42 to 44: Change to: “However, the arrival of exotic species may impose new challenges for native species, like new competitors and diseases, predation and habitat alteration, among others”. Monoculture and economic damage are not included in the “dispersal of organisms” since it is a human alteration done with intention. Besides, economy is not important for native species, it is only important for humans.
R: modified
Line 48: Remove the second “as” after spiders and before have. Add a comma after “spiders”.
R: removed
Line 67: Change “incidental” to “museum”. Check the gap between the end of this sentence and “Our study”.
R: correcteds
Line 73: Change “exhaustive” to “extensive”.
R: corrected
Methods:
Line 93: There is a “3” at the end of this line that I imagine that was for the title of the Results.
R: corrected
Reviewer 3 Report
Comments and Suggestions for Authors
Overall, an interesting paper that only needs minor revision. There were some spelling errors and a couple of the figures should be retaken. However, once resolved this will be suitable.
Line 13: locations to "location"
Line 14: locations to "areas"
Line 14: mean to "means"
Line 16: This invasive species to "Invasive species"
Line 17-18: It is wrong to state spiders have not been extensively studied as invasive species, whole book chapters and countless papers have been written! Delete.
Line 18: Rephrase to say that arachnids have not been extensively studied in Panama in terms of invasive species.
Line 20: field collection to "conducting fieldwork"
Line 22: "potential effects" to what? habitat? indigenous/endemic species?
Line 26-28: "A taxa for 26 which there is less information are arachnids, and this is the case of reports on invasive species in 27 Panama." you need to reprhase this sentence for reason given above.
Line 48: "non-araneid" would refer only to Araneidae! delete and just leave as "invertebrates, including spiders"
Line 55: enough to "continued" (continued attention)
Line 69: and it did not include to "but it did not include"
Line 82: haphazardly to "oppurtunisticly"
Line 86: delete "the"
Line 93: Delete 3.
Line 98: Add bracket to (4 spp.)
Line 101: Italicise scientific name
Line 151: contrasting with to "in comparison to"
Line 188: Itaclicise name.
Fig. 4: Photograph of the epigyne can be improved and made at higher magnification.
Fig. 6: Male palp is taken at a funny angle, those are not the true prolateral and retrolateral views. Please check recently published photographs of this species and replicate their views by orientating the femur
Author Response
Reviewer 3
Overall, an interesting paper that only needs minor revision. There were some spelling errors and a couple of the figures should be retaken. However, once resolved this will be suitable.
Line 13: locations to "location"
R: corrected
Line 14: locations to "areas"
R: corrected
Line 14: mean to "means"
R: corrected
Line 16: This invasive species to "Invasive species"
R: corrected
Line 17-18: It is wrong to state spiders have not been extensively studied as invasive species, whole book chapters and countless papers have been written! Delete.
R: We modified this sentence.
Line 18: Rephrase to say that arachnids have not been extensively studied in Panama in terms of invasive species.
R: modified
Line 20: field collection to "conducting fieldwork"
R: corrected
Line 22: "potential effects" to what? habitat? indigenous/endemic species?
R: modified
Line 26-28: "A taxa for 26 which there is less information are arachnids, and this is the case of reports on invasive species in 27 Panama." you need to reprhase this sentence for reason given above.
R: modified
Line 48: "non-araneid" would refer only to Araneidae! delete and just leave as "invertebrates, including spiders"
R: modified
Line 55: enough to "continued" (continued attention)
R: modified
Line 69: and it did not include to "but it did not include"
R: modified
Line 82: haphazardly to "oppurtunisticly"
R: modified
Line 86: delete "the"
R: deleted
Line 93: Delete 3.
R: deleted
Line 98: Add bracket to (4 spp.)
R: added
Line 101: Italicise scientific name
R: modified
Line 151: contrasting with to "in comparison to"
R: modified
Line 188: Itaclicise name.
R: corrected
Fig. 4: Photograph of the epigyne can be improved and made at higher magnification.
R: corrected
Fig. 6: Male palp is taken at a funny angle, those are not the true prolateral and retrolateral views. Please check recently published photographs of this species and replicate their views by orientating the femur.
R: corrected
Round 2
Reviewer 2 Report
Comments and Suggestions for Authors
No further comments to add. The authors add all my suggestions and I believed the paper was improved.